# Development of Broad-Spectrum Antiviral Agents—Inspiration from Immunomodulatory Natural Products

**DOI:** 10.3390/v13071257

**Published:** 2021-06-28

**Authors:** Mengxun Zhang, Jiaqing Zhong, Yongai Xiong, Xun Song, Chenyang Li, Zhendan He

**Affiliations:** 1Department of Pharmacy, School of Medicine, Health Science Center, Shenzhen University, Shenzhen 518060, China; zhangmx@szu.edu.cn (M.Z.); 2017153037@email.szu.edu.cn (J.Z.); xsong@szu.edu.cn (X.S.); 2College of Physics and Optoelectronic Engineering, Shenzhen University, Shenzhen 518060, China; 3College of Pharmacy, Zunyi Medical University, Zunyi 563000, China; yongaixiong@szu.edu.cn; 4College of Pharmacy, Shenzhen Technology University, Shenzhen 518118, China

**Keywords:** antiviral, natural products, immunomodulatory

## Abstract

Developing broad-spectrum antiviral drugs remains an important issue as viral infections continue to threaten public health. Host-directed therapy is a method that focuses on potential targets in host cells or the body, instead of viral proteins. Its antiviral effects are achieved by disturbing the life cycles of pathogens or modulating immunity. In this review, we focus on the development of broad-spectrum antiviral drugs that enhance the immune response. Some natural products present antiviral effects mediated by enhancing immunity, and their structures and mechanisms are summarized here. Natural products with immunomodulatory effects are also discussed, although their antiviral effects remain unknown. Given the power of immunity and the feasibility of host-directed therapy, we argue that both of these categories of natural products provide clues that may be beneficial for the discovery of broad-spectrum antiviral drugs.

## 1. Introduction

The battle between humans and viral infections has a long history and continues today, with the main tools available to humans being vaccines and antiviral drugs. Some viral infections have been eliminated by inoculation with vaccines. Many successful antiviral drugs, such as oseltamivir and sofosbuvir, have been developed, bringing significant benefits for patients. However, some viral infections are more difficult to cure and prevent. For example, HIV, Ebola, Zika, hepatitis B, and COVID-19 remain problems worldwide. Effective vaccines and powerful drugs are still not available for some pathogens. Moreover, the currently available vaccines and antiviral drugs are threatened by virus mutations. Therefore, the development of new antiviral drugs is necessary.

Current antiviral drugs are usually designed to target distinct virus groups, which can make them ineffective in treating emerging viruses. Developing broad-spectrum antiviral drugs could be a way to solve this problem. Host-directed therapy [1] is an approach to developing broad-spectrum antiviral drugs that includes two strategies: (I) interfering with host cell mechanisms, for example, by targeting common host factors used for viral entry or viral replication, and (II) enhancing the immune responses mounted against pathogens and reducing inflammation. Significant efforts have been made in this field during recent years. Some compounds and natural products with broad-spectrum antiviral activity have been discovered, and the related mechanisms and potential targets in host cells have been explored, providing clues for drug discovery. 

This review describes strategies for developing broad-spectrum antiviral drugs, with an emphasis on the treatment of viral infections by enhancing immunity. We summarize the natural products and extracts of herbs that show antiviral activity by enhancing the immune response. Chemical entities that enhance immunity are also summarized, and their potential use in the treatment of viral infections is discussed.

### 1.1. Viral Infections and Current Treatment

There are 219 viral species recognized as human pathogens [2]. Influenza is one of the most widely known viral infections, and periodic influenza pandemics resulting from antigenic variation have led to thousands of deaths [3]. Coronaviruses have been a major public concern in recent years due to three outbreaks: severe acute respiratory syndrome coronavirus (SARS-CoV) in 2003, the Middle East respiratory coronavirus (MERS-CoV) in 2012, and COVID-19 in 2019. Ebola virus disease (EVD) is also a severe illness that threatens public health, for which there are limited treatment methods and no effective vaccine, leading to repeated outbreaks and high fatality rates in Africa [4,5]. Besides, chronic viral infectious diseases also affect human beings. According to the WHO, there are 38 million people living with HIV [6]. Chronic viral hepatitis induced by the hepatitis B and C viruses affects 325 million people worldwide and may lead to cirrhosis, cancer, and even death [7].

Great efforts have been made to prevent and treat viral infections. Vaccines are powerful tools for inhibiting viral infections by harnessing adaptive immunity. Some viral pathogens, such as smallpox virus, have been eradicated by vaccination [8]. However, the effectiveness of vaccines is limited by viral evolution (e.g., that of influenza and varicella-zoster viruses). The methods used for vaccine preparation are based on the knowledge of pathogens and therefore lag behind the emergence of new viral diseases. Antiviral drugs are another important tool for the treatment of viral infections, and around 90 new antiviral drugs have been approved since 1963. However, all of these drugs are primarily effective against only nine human viral pathogens: HIV, HBV, HCV, influenza virus, respiratory syncytial virus, human papillomavirus, human cytomegalovirus, herpes simplex virus, and varicella-zoster virus [9]. This indicates that effective drugs are still lacking for many viral infections. The currently approved drugs mainly target specific processes in the viral life cycle; for example, there are inhibitors of viral DNA or RNA synthesis (acyclovir and zidovudine), viral protease inhibitors (saquinavir and ritonavir), and viral neuraminidase inhibitors (zanamivir and oseltamivir). This target-specific strategy makes drugs accurate and effective for specific viral species. However, it also limits the applicability of these drugs, making them ineffective for treating new viruses. The lack of effective drugs for other viral infectious diseases and newly emerging viruses highlights the importance of developing broad-spectrum antiviral drugs.

### 1.2. Strategies for Developing Broad-Spectrum Antiviral Drugs

By studying the intricacies of host–pathogen interactions, scientists discovered that pathogens can be inhibited by targeting host cell proteins and functions. [10] Theoretically, this strategy could yield candidates with broader applicability, since the target proteins do not relate to specific viruses. In recent decades, some reasonable host cell targets and promising compounds have been found. Most of these host cell proteins are involved in the viral life cycle [11,12]. For example, statins (Figure 1) were found to inhibit HIV by targeting HMG-CoA reductase in host cells [13], and cyclosporine A (Figure 1) fights against several types of viruses, including herpes simplex virus (HSV) [14], vaccinia virus (VV) [15,16], vesicular stomatitis virus (VSV) [17], severe acute respiratory syndrome coronavirus (SARS-CoV) [18], and influenza virus [19], by targeting cellular cyclophilins to abrogate viral replication. 

In addition to targeting host cell proteins to interfere with the viral life cycle, modulating the host’s immune response is another strategy for developing broad-spectrum antiviral drugs. The most successful case in this field is recombinant type Ⅰ interferons (IFNs), which can be used for the treatment of chronic hepatitis B and C and cutaneous herpes and warts induced by virial infections. Some chemical compounds can also enhance immunity; for example, imiquimod (Figure 1) can activate Toll-like receptor 7 (TLR7) and induce the secretion of various cytokines, including IFN-α, IL-1, IL-6, and TNF, which finally leads to antiviral effects [20]. The strategy of modulating the immune response has some obvious advantages: candidates can be developed without full knowledge of pathogens, and there is a lower incidence of drug resistance induced by viral mutation.

## 2. Fighting Viruses by Modulating Immunity

### 2.1. Immune Response during Viral Infection

As a result of millions of years of evolution, the immune system has become a powerful preventer of viral infection (Figure 2). Innate immunity provides the first line of host defense in a non-specific manner. This defense is mediated by protective tissue (e.g., skin and mucosa), natural immuno-molecules (e.g., complement, cytokines, and lysozymes), and immune cells (e.g., natural killer (NK) cells, macrophages, and dendritic cells (DCs)). Taking influenza as an example, pathogen-associated molecular patterns (PAMPs) are recognized by various pattern recognition receptors (PPRs) on mucosal epithelial cells and DCs, which leads to the secretion of cytokines and chemokines. Type I IFNs are cytokines important in antiviral defense because they can modulate the expression of hundreds of genes to produce widespread effects, such as inducing an antiviral state in infected cells, promoting macrophage and NK cell functions, and activating adaptive immunity [21,22]. Besides mucosal epithelial cells and DCs, the complement system can also sense danger signals, leading to the initiation of a serine protease cascade. The effects of this cascade are lysis, opsonization, and chemotaxis [23,24,25]. While innate immunity provides a rapid, non-specific response to a virus, adaptive immunity fights pathogens more specifically. After sensing antigens directly on circulating pathogens or indirectly on antigen-presenting cells (APCs), sets of gene segments are rearranged and assembled in T and B lymphocytes to produce specific antigen receptors and antibodies to kill infected cells and neutralize the virus [26].

### 2.2. Antiviral Effects of Immunomodulatory Natural Products

Natural products are an important source of drug discoveries [27]. Many herbs have been used to prevent and treat viral infections since ancient times, and their active antiviral ingredients have been widely studied [28,29]. Therefore, the screening of natural products for drug candidates that enhance immunity, inhibiting viruses, is a promising avenue of research. Herein, we describe natural antiviral products that work by enhancing immunity based on their chemical categories (Table 1).

Saponins are important bioactive natural products that consist of an aglycone and oligosaccharides. Several saponins exhibit antiviral activity by enhancing immunity (Figure 3). Astragaloside IV is a saponin isolated from *Radix astragali*, which can enhance LPS- or Con-A-induced T and B lymphocyte proliferation and antibody production [30]. In 2006, Zhang et al., found that astragaloside IV can significantly reduce the morbidity and mortality for coxsackievirus B3 (CBV3)-induced myocarditis in vitro and in vivo. The higher levels of IFN-γ mRNA in treated mice indicated that astragaloside IV may exert antiviral effects by upregulating IFN-γ [31]. In 2015, Gui et al., found that astragaloside IV inhibits NF-κB signaling by increasing A20 (TNFAIP3) expression, leading to the alleviation of cardiac inflammation in coxsackievirus-B3-induced myocarditis [32]. Saikosaponins A and D (Figure 3) are saponins that were isolated from *Bupleurum falcatum* L. in 1968 [33], and their immunomodulatory effect was first evaluated in 1985, when Yamaguchi et al., pretreated BALB/c mice with saikosaponins A and D and a hemolytic plaque test indicated that both compounds can increase humoral immunity against LPS and decrease humoral immunity against heterologous erythrocytes [34]. In 1988, Yoshio et al., reported that the intraperitoneal administration of saikosaponins A and D leads to cell accumulation in the abdominal cavity. Moreover, saikosaponin D significantly activated peritoneal macrophages, which showed enhanced phagocytic and cytostatic activity and a higher number of cellular lysosomal enzymes and surface antigens [35]. The antiviral activity of these saikosaponins was found to be related to the enhancement of the immune system. Saikosaponin A exhibited protective effects against influenza A in vivo and in vitro. In B6 mice infected with influenza virus PR8, saikosaponin A selectively modulated neutrophil and monocyte recruitment in the lung early in the innate immune response and led to decreased morbidity and mortality in the mice. The protective effects were mediated by downregulating the NF-κB signaling pathway and inhibiting caspase 3 activation, which are related to the secretion of proinflammatory cytokines and the nuclear export of viral proteins, respectively [36]. In 2017, Yang et al., found that saikosaponins A and D can alleviate PCV2-induced pathological damage in mice, and the protective effects were at least partially due to their immunoregulatory effects. Compared with the PCV2 control group, treatment with saikosaponins A and D increased IgG and white blood cell (WBC) levels in the serum [37]. Hu’s research showed that saikosaponin A and D treatment upregulates IL-2 expression and increases IFN-γ levels in endothelial cells challenged with PPRSV [38]. Ginsenoside Rb1 (Figure 3) is a type of triterpenoid saponin isolated from *Panax quinquefolius* L. that was first identified by Ohsawa in 1966 [39]. Recently, Yang found that ginsenoside Rb1 has significant antiviral effects against enterovirus 71 (EV71). The administration of ginsenoside Rb1 protected mice from paralysis and death induced by EV71, reduced histological damage in muscles of the limbs, and decreased the level of viral RNA. Further experiments demonstrated that the antiviral mechanism of Rb1 enhances immunity. Ginsenoside Rb1 treatment promoted the secretion of various cytokines, including TNF-α, IL-10, and IFN-β and γ, and increased the levels of antibodies in the serum [40]. 

Celastrol (Figure 4) is a terpenoid isolated from *Tripteryguim regelii* (Celastraceae) that has been identified as having antiviral activity toward SARS-CoV [41], HIV [42], and DENV [43]. The antiviral mechanisms include both directly interacting with viral proteins and modulating immunity. In Yu’s research, celastrol inhibited the replication of DENV in Huh-7 cells and protected DENV-infected mice from death and paralysis. Further investigation showed that celastrol activates the JAK/STAT signaling pathway, which leads to the expression of IFN-α, initiating an antiviral state in host cells [43].

Phenylpropanoids are secondary metabolites comprising one or several phenylpropane (C6–C3) units. The antiviral activities of phenylpropanoids have been widely studied. Our group found that phenylethanoid glycosides isolated from *Ligustrum purpurascens* Y.C. Yang (Oleaceae) inhibit H1N1 in vitro and in vivo. In splenocytes, treatment with the glycosides induced the secretion of IFN-γ and altered the CD4+/CD8+ cell ratio. Moreover, the protective effects of the glycosides were abrogated by the knockout of the IFN-γ gene in mice [72]. After bioactivity-directed isolation, we found that acteoside (Figure 5) is the major bioactive natural product from the mixture. Acteoside showed anti-VSV and anti-influenza effects in vitro and in vivo. The mechanism of IFN-γ upregulation was explored, and it was found that acteoside activates T-bet in lymphocytes and promotes the transcription of the IFN-γ gene. Acteoside also promoted the phosphorylation of extracellular regulated protein kinase (ERK), which led to the proliferation of primary lymphocytes [44]. Salidroside (Figure 5), one of the main bioactive components of *Rhodiola rosea* L. (Crassulaceae), is a phenylethanoid glycoside effective in treating coxsackievirus B3 (CBV3) [45], dengue virus (DENV) [46], and respiratory syncytial virus (RSV) [47] that acts by enhancing immunity. According to Wang, salidroside significantly reduced mortality from myocarditis in CVB3-infected mice, decreased the virus titers of infected cells, and reduced injury according to enzyme levels in the serum. RT-PCR showed that treatment with salidroside increases the expression of IL-10 and IFN-γ in heart tissue, while it decreased the expression of TNF-α and IL-2, indicating that its antiviral effects may be due to enhanced host immunity and reduced inflammation [45]. In 2016, Sharma et al., evaluated salidroside for its efficacy against DENV [46]. Salidroside treatment decreased the expression of the viral envelope protein and intracellular viral load in infected THP-1 cells. Analysis with ELISA and immunoblots showed that salidroside enhances the expression of retinoic acid-induced gene I (RIG-I) in host cells and then upregulates downstream signaling pathways involving interferon regulatory factors (IRF3 and IRF7) and IFN-α. The treatment of human peripheral blood mononuclear cells (hPBMCs) with salidroside increased the number of NK cells and CD8^+^ T cells, which may also contribute to the antiviral effect [46]. Arctigenin (Figure 5) is a lignanolide found in *Arctium lappa* L. that effectively treats several viral infections, including H1N1 [48], porcine circovirus type 2 (PCV2) [49], Japanese encephalitis virus (JEV) [50], HIV [51], and HSV [52]. In some cases, arctigenin has been found to interfere with specific processes in the viral life cycle [48,51,52], while in others, it has been shown to exert immunomodulatory effects. In 2018, Tong et al., found that arctigenin activates macrophages and increases the secretion of TNF-α and TGF-β1 and that the activation of the TLR6/NOX2/MAPK signaling pathway is responsible for these effects [53]. Arctiin (Figure 5) is the aglycone of arctigenin. According to Uesato, arctiin exhibits a significant protective effect against influenza A virus in both normal mice and mice with 5-fluorouracil-induced immunosuppression. Notably, arctiin induced more virus-specific antibodies than those observed for the blank control and oseltamivir groups [48]. 4-Methoxycinnamaldehyde (Figure 5) is an important volatile constituent of *Agastache rugosa*, a traditional medicine in China that has been used to treat infectious diseases for thousands of years. In 2009, Lin reported that it can inhibit RSV, with an EC_50_ of 0.055 μg/mL. Further experiments showed that 4-methoxycinnamaldehyde not only prevents viral entry but increases the secretion of IFN in a human larynx carcinoma cell line [54].

Anthraquinones in *Aloe* sp. exhibit antiviral activity by modulating immunity. Aloin (Figure 6) is the major anthraquinone glycoside extracted from *Aloe*. In 2019, Huang et al., found that aloin is effective against influenza in vitro and in vivo, and the mechanisms responsible include the direct inhibition of neuraminidase activity and immune response modulation. In a neuraminidase (NA) assay, aloin showed an inhibitory effect against both a normal strain and an oseltamivir-resistant strain. Meanwhile, an animal study showed that aloin treatment increases the number of antigen-specific T cells in the lung. The combined use of aloin and oseltamivir had a synergistic effect. Cytokine staining showed that aloin increases the phosphorylation of STAT and promotes T-bet and IFN-γ, which may be responsible for the synergistic effect [55]. Aloe-emodin (Figure 6) can be isolated from *Aloe* or rhubarb. In 2008, Lin et al., reported that aloe-emodin fights JEV and EV71 by inducing IFN-α. Dual-luciferase reporter assays revealed that aloe-emodin significantly increases the promoter activity of interferon-stimulated response element (pISRE)-Luc and gamma-activated sequence (pGAS)-Luc by 2.14- and 2.03-fold, respectively [56].

Flavones have long been known for their antiviral activities, and the possible mechanisms responsible include both interfering with the viral life cycle directly and modulating immunity. Baicalin (Figure 7) is a flavonoid isolated from *Scutellaria baicalensis* Georgi that exhibits inhibitory activity against influenza viruses [57,58], possibly through the inhibition of neuraminidases [58] and by enhancing immunity [59]. It was observed that baicalin increases IFN production in H1N1-infected mice, indicating that the antiviral effect is related to immune system enhancement, and the lack of protective effects observed in IFN-deficient or IFN-receptor-deficient mice supported this relationship. Western blotting showed that baicalin promotes the secretion of IFN-γ in T cells and NK cells, leading to the upregulation of the downstream pathway [59]. In 2015, Cho et al., reported that aqueous extracts from *Epimedium koreanum* show an inhibitory effect against several viral pathogens, including HSV, influenza A virus, Newcastle disease virus (NDV), and VSV. The EC_50_ values ranged from 0.28 to 0.94 μg/mL in HEK293T and RAW264.7 cells infected with different viruses. The efficacy of the extracts in BALB/c mice was also identified using different influenza subtypes. The broad-spectrum antiviral properties indicated that the effects may derive from host cells instead of viral pathogens. Mechanistic studies showed that the extracts induce the secretion of type I IFN and IL-6, inducing an antiviral state in the cells. Quercetin (Figure 5) was identified in the extracts as the main active component [60]. In 2016, Kim et al., reported that quercetin effectively reduces influenza A viral infection in MDCK cells. Proteomic analysis showed that 56 proteins are modulated by quercetin. Proteins related to viral replication (e.g., FN1) were downregulated, while others (e.g., CCT4) were upregulated, acetylated, or phosphorylated [61]. Vitexin (Figure 7) is a flavonoid isolated from the dried flowers of *Trollius chinensis* and is effective against influenza A in vitro. RT-PCR and Western blotting showed that vitexin upregulates TLR4, and ELISA showed that it increases the level of IFN-β and decreases the level of TNF-α in the early stages of infection [62].

Matrine and oxymatrine (Figure 8) are alkaloids that isolated from *Sophora alopecuraides* L. In research by Li’s group, matrine exhibited a protective effect against a co-infection of PRRSV and PCV2 in mice. It also alleviated the damage to organs and decreased the copy number of the PCV2 gene. Further investigation showed that matrine enhances phagocytosis by peritoneal macrophages and increases lymphocyte proliferation [63]. Oxymatrine (Figure 8), the natural oxidized form of matrine, possesses strong immunomodulatory properties. In Wang’s research, lymphocytes were isolated from chronic hepatitis B patients and treated with oxymatrine, after which the upregulation of the TLR9 signaling pathway was observed, which led to increased levels of type I interferon and TNF-α. These results suggest that oxymatrine may inhibit HBV by enhancing the host’s immune response [64]. These immunostimulatory effects were also observed in other research; for example, Liu et al., showed that oxymatrine promotes the maturation of dendritic cells [65], and Ye et al., found that it encourages CD8+ T cells isolated from tumor-bearing mice to secrete IFN-γ, TNF-α, and IL-2, which may be responsible for the synergistic anticancer effect of oxymatrine and cisplatin [66]. Trolline is an aromatic alkaloid isolated from *Trollius chinensis* that has been shown to upregulate TLR4 and increase the level of IFN-β, which has a protective effect against influenza virus [62].

Polysaccharides isolated from *Glycyrrhiza uralensis* exhibit a variety of biological activities (e.g., antioxidant and immunomodulatory). In 2020, Wang et al., studied a glycyrrhiza polysaccharide (GP). HPLC, CD, and FT-IR showed that GP is an acid glycoprotein with a complex structure. Moreover, Wang et al., found that GP can protect MDBK cells from infections of bovine viral diarrhea virus (BVDB), and further experiments showed that GP does not inhibit the replication of the virus but promotes the expression of IRF-1 and IRF-3, which are important regulatory factors for interferon. These results indicate that the antiviral activity of GP is at least partially due to the modulation of the immune response [67].

*Radix isatidis* is traditionally used to treat viral and bacterial infections in China. *Radix Isatidis* polysaccharide (RIP) is an effective treatment for several viruses, including HSV-2 [68], influenza A virus [69], and HBV [70]. According to Wang, its antiviral activity may be due to the enhancement of the immune response. In HepG2.2.15 cells infected with HBV, RIP treatment decreased the levels of both HBsAg and HBeAg. Results from ELISA showed that RIP upregulates IFN-α in cells and activates the JAK/STAT signaling pathway by promoting the phosphorylation of related kinases, which leads to a protective effect against HBV [70].

An acidic polysaccharide (APS) isolated from *Cordyceps militaris* grown on germinated soybeans was also found to have antiviral activity by modulating immunity. HPLC analysis, along with a hydrolysis and methylation experiment, showed that APS is a polysaccharide with a molecular weight of 576 kDa and is mainly composed of Araf–Araf–Galp–GalAp residues. In mice infected with influenza A, the administration of APS increased the survival rate and decreased virus titers in bronchoalveolar lavage fluid and the lungs. The higher levels of TNF-α and IFN-γ compared with the control group indicated that the antiviral effects may be a result of immune system enhancement. In RAW264.7 murine macrophage cells, APS enhanced the levels of proinflammatory cytokines and signaling molecules, such as IL-1β, IL-10, IL-6, TNF-α, and nitric oxide (NO) [71].

## 3. Natural Products That Enhance Immunity

The modes of immunomodulation of the natural products discussed in Section 2 are varied, but all lead to protective effects against viral infections. Some natural products activate immune cells, such as T cells, B cells, macrophages, and neutrophils, while others affect cytokines, such as IFN. Meanwhile, some natural products modulate the expression of surface molecules, such as CD86 and MHC-II. The signaling pathways modulated by these natural products are different and include the IFN, TLR, and JAK/STAT signaling pathways. Based on these observations, we propose that the discovery of antiviral agents from immunomodulatory natural products is possible, especially from immunoenhancing species. Here, we summarize immunoenhancing natural products divided into three categories: small molecules, proteins/peptides, and polysaccharides.

### 3.1. Small Molecules

A number of small molecules with antiviral activity that enhance immunity have been identified. As such, we suggest that further broad-spectrum antiviral agents could be derived from immunomodulatory natural products for which antiviral activity has not yet been explored. Herein, we describe these immunomodulatory small molecules based on their chemical categories (Table 2).

Many saponins have been reported with immunomodulatory activities. In a study by Yang, saponins isolated from *Astragalus membranaceus* (Fisch.) Bge. enhanced splenocyte proliferation induced by several antigens. In OVA-immunized mice, treatment with saponins led to higher levels of antigen-specific antibodies in the serum [73]. Pure compounds isolated from total saponins have been studied. Astragaloside II (Figure 9) significantly restored the proliferation of splenic T cells in immunosuppressed mice and increased the levels of IFN-γ and IL-2. The treatment of CD4+ T cells with astragaloside II promoted the expression of activation markers, such as CD25 and CD69. Competitive experiments showed that the immunomodulatory effects of astragaloside II were mediated by CD45 [74]. In 2005, Ohmoto et al., reported that astragaloside VII (Figure 9) significantly increases LPS- and PHA-induced IL-2 secretion in heparinized peripheral whole blood [75]. In 2012, Nalbantsoy et al., found that astragaloside VII and macrophyllosaponin B increase the LPS induction of IL-2 and IFN-γ in mice, and specific antibodies for astragaloside VII and macrophyllosaponin B were detected by ELISA. Mitogen-induced splenocyte proliferation was also promoted by these two compounds. A reporter gene assay showed that they have no effect on NF-κB or NAG-1 activity, indicating that astragaloside VII and macrophyllosaponin B enhance the immune response without stimulating inflammatory cytokines in mice [76]. Saikosaponins are major bioactive ingredients derived from *Bupleurum chinense* DC. Saikosaponins B1, B2, and D (Figure 8) were found to stimulate prostaglandin E2 biosynthesis in rat peritoneal macrophages [77]. Saikosaponin B2 treatment also increased phagocytosis in vitro [78]. (−)-β-Sitosterol-3-*O*-β-D-(6-*O*-palmitoyl)glucopyranoside (Figure 9) is a phytosterol isolated from *Phyllanthus songboiensis*. In 2015, Ren et al., found that the treatment of NK cells with (−)-β-sitosterol-3-*O*-β-D-(6-*O*-palmitoyl)glucopyranoside in the presence of IL-12 promotes the secretion of IFN-γ. The production of IFN-γ in a combinational treatment group was significantly higher than with IL-12 treatment alone [79]. In 1998, Yoshikawa et al., found that a saponin mixture from the white seeds of *Lablab purpureus* shows potent adjuvant activity. Four saponins were identified in this fraction: lablabosides A, B, and C, and chikusetsusaponin IVa (Figure 9) [80]. 

Some terpenoids also show the ability of modulating the immune system. Glycyrrhetinic acid (GA, Figure 10) is a terpenoid isolated from *Glycyrrhiza uralensis* Fisch. In 2011, Peng et al., explored the influence of GA on the TLR signaling pathway. The murine macrophage cell line Ana-1 was treated with GA, and the expression of different subtypes of TLR was tested by qRT-PCR. TLR4 was found to be significantly upregulated by GA. Further experiments showed that downstream signaling molecules, including MyD88, IL-6, and IFN-β, are also upregulated [81]. The immunomodulatory activity of liposomes of GA was studied by Zhao et al., T and B lymphocytes were treated with the liposomes separately in vitro; enhanced proliferation was observed with or without a co-stimulator (LPS or PHA), and there was a higher level of antibodies [82]. Pentalinonsterol (Figure 10) was first isolated from the roots of *Pentalinon andrieuxii* in 2012, and it was identified as having antiparasitic activity [83]. The immunomodulatory properties of PEN were explored in 2017. Treatment with pentalinonsterol activated macrophages and bone-marrow-derived dendritic cells (BMDCs). In addition, increased expression of transcription factors (NF-κB and AP1), proinflammatory cytokines, and surface molecules related to antigen presentation was observed. In mice immunized with OVA, the administration of pentalinonsterol increased the level of antibodies in the serum; meanwhile, splenocytes and lymph node cells isolated from treated mice secreted more cytokines, such as IFN-γ and IL-10, when compared with the control group [111]. In a study by Gupta, pentalinonsterol treatment to combat parasites was found to induce T cell proliferation and cytokine release [84]. The immunostimulatory effects of *Andrographis paniculata* extract and andrographolide (Figure 9) were tested in vitro and in vivo. It was found that both the extract and andrographolide enhance the formation of specific cytotoxic T lymphocytes. The addition of the extract or andrographolide to splenocytes inhibited EL4 thymoma cells and strengthened the effector T cells, leading to longer survival when administered to EL4-bearing mice [85]. 14-Deoxy-11,12-didehydroandrographolide (Figure 10) is also a terpenoid isolated from *Andrographis paniculate*, and it was found to enhance the innate immunity of intestinal epithelial cells. The treatment of an HCT-166 cell line with 14-deoxy-11,12-didehydroandrographolide led to increased expression of the antimicrobial protein hBD-2. Western blotting showed that the activation of p38 may be related to immune enhancement, which was supported by the observation that p38 inhibitors block the effects of the natural product [86]. In 2015, two diterpenoids isolated from *Cinnamomum cassia* Presl (Lauraceae)—cassiabudanols A and B (Figure 10)—were reported to exhibit immunostimulatory activity. The treatment of splenocytes with cassiabudanols A and B could enhance LPS-induced cell proliferation. Further experiments showed that cassiabudanol B also increases the ratio of CD4+ and CD8+ T cells in splenocytes [87]. Cinnamomols A and B (Figure 10) are diterpenoids isolated from *C. cassia*, and both exhibit significant immunostimulatory activity. In a Con-A-induced splenocyte proliferation assay, cinnamomols A and B enhanced the proliferation of murine T cells at concentrations of 0.391 to 100 μM. Fluorescence-activated cell sorting (FACS) analysis showed that cinnamomol A promotes the differentiation of CD4+ T cells, while it had no effect on CD8+ T cells [88]. Lycopene (Figure 10) is a natural product widely found in red fruits and vegetables. Xu et al., showed that lycopene can alleviate aflatoxin B1-induced immunosuppression [89]. Treatment with lycopene relieved injury of the spleen induced by aflatoxin B1, increased the relative proportion of CD4+ and CD8+ T cells, and increased the levels of IL-2, TNF-α, and IFN-γ in the serum and spleen. Studies on the mechanism responsible showed that the protective effect is due to decreased oxidative stress and the inhibition of splenocyte apoptosis [89]. Astaxanthin (Figure 10) is a carotenoid found in shrimp, crabs, algae, and other marine organisms. The immunostimulatory activity of astaxanthin has been widely explored. A plaque formation assay showed that the treatment of normal mice with astaxanthin can enhance the production of antibodies against sheep red blood cells. The depletion of T cells from splenocytes blocked the effects of astaxanthin, indicating that it may work by promoting antigen presentation by T helper cells [90,91]. In a study by Park, dietary astaxanthin stimulated mitogen-induced lymphoproliferation, enhanced the activity of NK cells, and increased the population of CD3+ T cells and B cells. Higher tuberculin responses and levels of plasma IFN-γ and IL-6 were also observed [92]. In mice made immunodeficient with cyclophosphamide, the administration of astaxanthin effectively protected the intestinal mucosa from damage. The development of Paneth cells decreased, while the expression of antimicrobial peptides and secretion of IgA significantly increased [93]. These immunomodulatory effects of astaxanthin illustrate its potential for use in treating infectious diseases. In fact, astaxanthin was found to reduce colonization levels and inflammation scores in BALB/c mice infected with *Helicobacter pylori* [94]. However, whether these effects are related to a modulation of immunity remains unknown.

Chlorogenic acid (Figure 11) is a phenylpropanoid isolated from *Flos lonicerae*. In 2004, Wu et al., found that the administration of chlorogenic acid promotes macrophage function in mice based on an assay measuring the clearance of charcoal particles. Chlorogenic acid can activate calcineurin in vitro, which was validated with both *p*-nitrophenyl phosphate and ^32^P-labeled RII peptide substrates, indicating that its effects in vivo may be mediated via the calcineurin/NF-ATc/IL-2 signaling pathways [95]. Phyllanthusmin C (Figure 11) is a lignan isolated from *Phyllanthus oligospermus*. In 2014, Deng et al., found that it promotes the secretion of IFN-γ by NK cells in vitro. Western blotting showed that treatment with phyllanthusmin C activates the p65 subunit of NF-κB, which then binds to the promoters of IFN genes. Competitive experiments showed that the blockade of TLR1 or TLR6 can diminish the effects of phyllanthusmin C, indicating that TLR/NF-κB is the disturbed signaling pathway [96].

Icariin (ICA; Figure 12) is an important bioactive monomer extracted from *Epimedium*. This flavonoid glycoside has been reported to modulate immune activity, raising the possibility that it could be used to treat viral infections. In some cases, icariin shows immunoenhancing effects, such as promoting Th1-lineage development and stimulating IgG production in mice [97] and inducing a notable expression of TLR9 in macrophages [98]. The anti-inflammatory activity of icariin has also been explored and may involve the downregulation of inflammatory cytokines [99,100], affecting T helper cells and regulatory T cells [101,102,103]. In 2015, Ajaghaku et al., found that extracts of *Millettia aboensis* (Hook F.) Baker (Leguminosae) exhibit both antioxidant and immune-enhancing properties. Further research showed that the butanol fraction is mainly responsible for this immunomodulatory activity. Bioactivity-directed isolation led to the discovery of rutin (Figure 12, quercetin-3-*O*-rutinoside). Compared to the DMSO control group, the rutin treatment of lymphocytes increased the population of CD4+ lymphocytes and promoted the secretion of IFN-γ [104].

Gonytolide A (Figure 13) is a dimeric chromanone that was isolated from the fungus *Gonytrichum* sp. in 2014. The natural product was discovered using a bioactivity-directed isolation assay based on an ex vivo *Drosophila* culture system for screening compounds that regulate the innate immune system. As a stimulator of the innate immune system, gonytolide A increases the production of IL-8 in human umbilical vein endothelial cells in vitro, indicating its potential for use against infectious diseases and tumors [105]. Structure-simplified gonytolide derivatives were synthesized in 2016, addressing the scarcity of gonytolide A. The synthesized analogues bischromone **1** and biflavone **2** both enhanced the innate immunity in *Drosophila* and mammalian cells [106]. 

Pyrrole alkaloids isolated from the fruits of *Morus alba* have been studied for their immunomodulatory activity. Two alkaloids—2-formyl-5-(hydroxymethyl)-1H-pyrrole-1-butanoic acid and 2-formyl-5-(methoxymethyl)-1H-pyrrole-1-butanoic acid (Figure 14)—have been found to significantly activate murine macrophages in vitro. The treatment of RAW264.7 cells with the two alkaloids increased the production of TNF-α, IL-12, and NO and promoted the phagocytosis of macrophages [107].

Organosulfur constituents in garlic have been studied by several researchers regarding their immunomodulatory activity. Alliin (Figure 15) was found to promote mitogen-induced cell proliferation in peripheral blood mononuclear cells (PBMCs) and to increase the secretion of IL-1β in PMBCs with or without LPS treatment [108]. Allicin (Figure 15) enhanced the production of proinflammatory cytokines in malaria-infected mice and increased the populations of macrophages, DCs, and CD4+ T cells, which prolonged the survival of infected mice [109]. In normal BALB/c mice, diallyl disulfide (Figure 15) increased the levels of WBCs and antibodies in the serum. In mice immunized with sheep red blood cell (SRBC), treatment with diallyl disulfide led to more effector B cells, according to the results of a plaque formation assay [110]. Ota et al., reported that all three compounds—alliin, allicin, and diallyl disulfide—enhance the secretion of IL-2, IL-4, and IFN-γ in intestinal Peyer’s patch cells. Furthermore, analyses of genetic profiles revealed the upregulation of 68–144 genes and the downregulation of 50–52 genes [112].

### 3.2. Peptides and Proteins

In recent years, peptides and proteins in natural products have been found to exhibit diverse biological activities. 

*Panax ginseng* C. A. Mey. is a traditional Chinese herb. The oligopeptide prepared from it has been investigated for its immunomodulatory activity in vivo. The intragastrical administration of the oligopeptide in BALB/c mice enhanced both cell-mediated and humoral immunity. Its influence on different types of immune cells was also tested, and it was found that it enhances macrophage phagocytosis and NK cell activity. Treatment with the oligopeptide also altered the population of T cells by increasing the relative proportion of CD4+CD8– T cells. Higher levels of IL-2 and IL-12 in the serum were also observed in oligopeptide-treated animals [113]. 

A polypeptide isolated from the fruit of *Cordyceps militaris* was also studied for its immunomodulatory activity. In immunosuppressed mice treated with cyclophosphamide, the administration of the polypeptide increased the spleen and thymus index, the white blood cell count, and the level of hemolysin in serum. Transcription analysis based on an mRNA microarray indicated that the polypeptide modulates the immune system through the transcription factors Ets1 and E2F [114].

LZ-8 is a polypeptide isolated from *Ganoderma lucidum*. It is a homodimer with a molecular weight of 24 kDa and contains 110 amino acids [115]. Yeh et al., found that LZ-8 activates murine peritoneal macrophages. ELISA showed that LZ-8 increases the production of IL-1β and IL-12p70. It also promoted the expression of MHC II on the surfaces of macrophages and enhanced the activation of T cells. These results indicate the ability of LZ-8 to activate innate and adaptive immunity [116]. The F3 glycoprotein can be isolated from *Ganoderma lucidum* and was first reported by Chien et al., The treatment of mononuclear cells isolated from umbilical cord blood with F3 increased the population of CD56^+^ NK cells and CD83+CD1a+ DCs. Using K562 tumor cells as a target, enhanced cytotoxicity for CD56^+^ NK cells was observed with F3 treatment [117,118].

In 2016, a water-soluble protein was extracted from *Panax quinquefolius* L. Electrophoresis and MALDI–TOF–MS suggested that the protein is a homodimer with a molecular weight of 31,086 Da. The treatment of murine peritoneal macrophages with this protein enhanced macrophage phagocytosis and facilitated the production of NO, TNF-α, and IL-6 [119]. 

Fraction 4 (F4) is a protein isolated from aged garlic, with a molecular weight of 11 kDa. Hirao et al., found that the treatment of murine macrophages with this protein enhances the phagocytosis of mast tumor cells [120]. Similarly, Morioka et al., found that F4 enhances the cytotoxicity of human peripheral blood lymphocytes against tumor cells, especially when F4 is used in combination with a suboptimal dosage of IL-2 or Con-A [121]. In 2010, three proteins (QR-1, QR-2, and QR-3) were isolated from garlic. These proteins can induce the proliferation of several types of immune cells, including splenocytes, thymocytes, and peripheral blood lymphocytes [122]. 

Aloctin A, a glycoprotein isolated from the leaves of *Aloe arborescens* Miller, exhibits various biological activities. Suzuki et al., showed that it has a mitogenic effect on lymphocytes and can activate complement by an alternative pathway [123]. In 1993, Imanishi et al., reported that the intravenous administration of aloctin A increases the cytotoxicity of the spleen and peritoneal exudate cells, indicating the activation of NK cells. The activation of T cells was also observed using a ^3^H-thymidine incorporation assay. Culturing T cells with aloctin A in combination with different types of accessory cell resulted in the production of cytokines, including IL-2, IL-3, and IFN-γ [124].

A glycoprotein isolated from the seeds of *Dolichos lablab* L. was shown to activate murine T lymphocytes. Its main properties are similar to those of concanavalin A, although some of the early events triggered by these two lectins differ, such as the initiation of IL-2 production [125].

FIP-fve is a 12.7 kDa glycoprotein isolated from *Flammulina velutipes*, comprising 114 amino acid residues [126]. It can elicit IFN-γ production in hPBMCs and hemagglutination [127]. The binding domain and targets of FIP-fve have been explored by site-directed mutagenesis and competition assays, which identified carbohydrate-binding modules (CBMs) at the C-terminal region of FIP-fve and glycans on the surface of hPBMCs as counterparts of the immunomodulatory effects [128]. 

### 3.3. Polysaccharides

Polysaccharides are important bioactive species, some of which have been used as drugs. For example, heparin has been used to treat thromboembolic diseases. Recently, DSTAT, a derivative of heparin, was found to inhibit CXCR4, and clinical trials related to oncology and COVID-19 are ongoing [129]. Herein, we summarize polysaccharides obtained from natural sources that show immunomodulatory effects, and their chemical details are provided in Table 3.

Verbascose is a galacto-oligosaccharide isolated from mung beans, *Phaseolus aureus*. Its immunomodulatory effects were evaluated in vitro and in vivo, and the treatment of RAW264.7 with the compound enhanced phagocytosis and promoted the release of NO and cytokines, including IL-6, IL-1β, IFN-α, and IFN-γ. The administration of verbascose in mice relieved the immunosuppression induced by cyclophosphamide, and recovery of the thymus and spleen index, hypersensitivity, hemolysin antibody, and lysozyme activity were observed [130].

Acemannan, a β-(1,4)-linked acetylated mannan derived from *Aloe vera*, has both antiviral and immunostimulatory properties. In CEM-SS cells, acemannan inhibited HIV in vitro, with an IC_50_ of 45 μg/mL [131]. When combined with either azidothymidine or acyclovir, acemannan acted synergistically to inhibit the replication of HIV-1 or HSV-1, respectively [132]. The administration of acemannan significantly improved the survival rate and quality of- life of felines with leukemia induced by oncornavirus infection [133]. However, whether these antiviral effects are related to immunomodulatory effects is not clear. In 1988, Womble et al., found that acemannan enhances the lymphocyte response to alloantigen in a dose-dependent manner. Further investigation found that the co-culture of T cells with monocytes pretreated with acemannan enhances the lectin response of the T cells, indicating that acemannan activates monocytes to enhance immunity. This conclusion was supported by the observation that treatment with acemannan increased IL-1 secretion in monocytes [134]. In 1992, Womble et al., reported that the acemannan treatment of primary splenocytes enhances the formation of cytotoxic T cells [135]. The effects of acemannan on DCs were also studied. Immature DCs were isolated from bone marrow and cultured with IL-4 and colony stimulating factor (CSF), and a phenotypic analysis and function test showed that the addition of acemannan promotes the maturation of DCs, which was identified through the higher expression of MHC-II, CD40, and CD45 on the surface and more potent stimulation of T cells [136]. These studies may indicate that the protective effects of acemannan in infectious diseases are partially due to immune system enhancement. MAP, a modified *Aloe* polysaccharide reported by Qiu et al., can increase the secretion of TNF-α from macrophages. In an immunosuppressed animal model induced by ultraviolet B, MAP showed a protective effect by inducing recovery from the hypersensitivity response in animals [137]. Aloeride, a polysaccharide with high molecular weight (Table 3) reported by Pugh et al., activated NF-κB in THP-1 human monocytic cells. Moreover, it was found to increase the levels of IL-1β and TNF-α, further illustrating its ability to activate macrophages [138]. 

Polysaccharides derived from *Polygonatum sibiricum* are immunostimulatory. In 2016, Yelithao et al., reported that two different polysaccharides, F1 (Mw: 103 kDa) and F2 (Mw: 628 kDa), can significantly increase the secretion of IL-1, IL-6, IL-10, IL-12, and NO from RAW264.7 cells [139]. In 2018, Zhao et al., found that a polysaccharide from *Polygonatum sibiricum* (PSP) can alleviate the immunosuppression induced by cyclophosphamide in BALB/c mice. Compared with the control group, treatment with PSP led to increased levels of IL-2 IL-8, and TNF-α in the serum and a decreased level of IL-10. The mitogen-induced proliferation of T and B cells was also restored by treatment with PSP in immunosuppressed mice [140].

In 2016, Wei et al., reported that RAP, a hyperbranched heteroglycan isolated from *Astragalus membranaceus*, with a molecular weight of 1334 kDa, can promote cytokine production in RAW264.7 cells via the TLR4 signaling pathway [141]. Wang et al., reported that an acid heteropolysaccharide (Mw = 5.6 kDa) from *Astragalus membranaceus* promotes the efficacy of a recombinant protein (rP-HSP90C) vaccine, leading to an increase in specific antibody titers in the serum [142].

In 2013, Zhao et al., reported a water-soluble low-molecular-weight polysaccharide (SCPP11) isolated from *Schisandra chinensis* (Turcz.) Baill, which was found to have immunostimulatory activity both in vivo and in vitro. The treatment of tumor-bearing mice with SCPP11 increased the thymus index and the levels of both IL-2 and TNF-α in the serum. In RAW 264.7 cells, SCPP11 enhanced the phagocytic activity and production of NO [143]. Further studies showed that the effects of SCPP11 can be significantly reduced by a TLR4-specific mAb, indicating that TLR4 may play an important role in this process [144]. In 2018, Yu et al., reported that the crude Schisandra polysaccharide (SCP) alleviates cyclophosphamide-induced immunosuppression at doses of 40–160 mg/kg. The leukocyte count, phagocytic function of macrophages, and levels of several cytokines can be promoted by SCP [145].

Polysaccharides are the major bioactive components in aqueous extracts of *Panax quinquefolium* L. Several polysaccharides have been prepared by different methods, and their immunological effects have been observed [146,147,148,149,150]. The polysaccharide CVT-E002 is a commercially available product that was observed to exhibit immunomodulatory activity. The treatment of splenocytes with CVT-E002 enhanced the proliferation of cells and the production of IL-2 and IFN-γ induced by Con-A, although CVT-E002 alone showed no activity [146]. GL-4IIb2 is a complex pectic polysaccharide isolated from *Panax quinquefolium* L. that can enhance Fc receptor expression on macrophages [148]. 

The immunomodulatory effects of polysaccharides isolated from *Bupleurum smithii* var. *parvifolium* have been explored in vivo and in vitro. According to Chen, this polysaccharide enhances the chemotaxis and phagocytosis of macrophages and increases the levels of proinflammatory cytokines and NO. However, for macrophages pretreated with LPS or complement, the polysaccharide slightly decreased the production of proinflammatory cytokines and NO. The mechanism behind this bidirectional modulation remains unclear [151,152]. An acid polysaccharide D3-S1 was isolated from the crude extract, and functional analysis showed that it inhibits complement activation by both classic and alternative pathways [153].

Polysaccharides isolated from *Prunella vulgaris* L. have been shown to enhance macrophage function. For example, PV2IV, a polysaccharide reported by Fang et al., stimulates the production of superoxide and NO in RAW264.7 cells as well as brain macrophage BV2 cells [154,155]. Polysaccharide P1 was isolated from the fruit of *Prunella vulgaris* L. and has been shown to increase the levels of TNF-α, IL-6, and NO in RAW264.7 cells. Antibody inhibition experiments showed that these effects may be related to TLR2, TLR4, or CR3 [156].

Hiroaki et al., reported that a crude polysaccharide fraction (AS-1) from the rhizome of *Anemarrhena asphodeloides* Bunge has immunomodulatory effects on intestinal Peyer’s patch cells. Aged BALB/c mice were treated with AS-1, and a sample of Peyer’s patch cells was taken for in vitro culture. ELISA showed that the AS-1 group had higher levels of IL-6, IL-10, and IFN-γ than the non-treatment group [157]. In mice immunized with OVA, treatment with AS-1 led to significant suppression of OVA-specific IgE. To illustrate the structure–activity relationship for this polysaccharide, mannanase digestion experiments were conducted and indicated that long hexosyl-oligosaccharides are important for the biological activity of AS-1 [157].

In 2019, the AAP70-1 polysaccharide was isolated from *Anemarrhena asphodeloides*. A structural analysis revealed that it is a branched polysaccharide with a molecular weight of 2.72 kDa. The treatment of RAW264.7 cells with AAP70-1 promoted phagocytic activity and the secretion of proinflammatory cytokines [158]. 

The immunomodulatory activity of polysaccharides isolated from *Morus alba* L. has also been studied. In 2000, Kim et al., reported that PMA, a polysaccharide isolated from the root bark of *M. alba*, enhances the proliferation of splenic lymphocytes in the presence of mitogens. However, a decrease in humoral immunity was also induced by PMA, indicating that PMA may promote the proliferation while inhibiting the differentiation of lymphocytes [159]. Polysaccharides isolated from the leaves of *Morus alba* were also studied. In weaning pigs, they improved the thymus and spleen index and increased the levels of antibodies and cytokines (IL-1β, IL-2, IL-8, and IFN-γ) in the serum [160]. 

In 2018, Bi et al., described a low-molecular-weight fucoidan (LMWF) isolated from *Undaria pinnatifida*. This polysaccharide activated RAW264.7 cells at concentrations ranging from 1 to 50 μg/mL, and greater cytokine (TNF-α and IL-6) secretion, NO release, and iNOS expression were observed. Western blots and competitive inhibitor assays showed that LMWF stimulates macrophages by activating NF-κB and MAPK signaling pathways [161].

PS-G is a polysaccharide isolated from *Ganoderma lucidum*. The treatment of murine macrophages with PS-G has been shown to promote the production of TNF-α and IL-12, and these effects could be abolished by the knockout of TLR4, indicating that TLR4 may be a putative receptor of PS-G [116].

The POP1 polysaccharide was extracted from the leaves of *Platycladus orientalis* L. Franco in 2016. The treatment of macrophages with POP1 significantly enhanced the expression and release of IL-6, IL-12, and TNF-α. Additionally, POP1 showed protective effects against HBV in HepG2 cells. POP1 also inhibited the expression of HBsAg and HBeAg and interfered with the replication of viral DNA in host cells. However, whether these antiviral effects are related to immune enhancement remains unclear [162].

A galactomannan isolated from *Antrodia cinnamomea* (ACP) was reported by Perera et al., and structural analysis showed that it contains an octasaccharide-repeating unit. The treatment of J774A.1 murine macrophages with ACP enhanced TNF-α production and phagocytosis. When cultured with *Escherichia coli*, the macrophages pretreated with APC showed more potent bactericidal activity compared with the control group [163].

In 2018, two polysaccharides, SHIP-1 and SHIP 2, were isolated from the fungus *Phellinus* sp. in culture. It was shown that the intragastric administration of SHIP-1 and SHIP-2 to normal mice can improve their immune response by increasing the levels of IL-2, IL-4, IFN-α, and IFN-γ in the serum [164].

BP-1 is a water-soluble polysaccharide that was isolated from *Hordeum vulgare* L. in 2019. In mice pretreated with BP-1, the immunosuppression induced by cytoxan was limited, and higher counts of white blood cells in the peripheral blood and bone marrow cells (BMCs) were observed compared with the control group. BP-1 also promoted the proliferation of solenocytes and enhanced the functions of NK cells and macrophages. The levels of IL-2, TNF-α, and IFN-γ in the serum were also higher in the BP-1 treatment group. Western blotting showed that BP-1 activates the TLR-4/TRAF6 signaling pathway in macrophages [165]. 

In 2019, Guo et al., reported the novel CCP polysaccharide isolated from *Craterellus cornucopioides*, and structural analysis showed that it has a triple-helix strand with a molecular weight of 1970 kDa. In vitro studies indicated that CCP can activate RAW264.7 cells and enhance their proliferation [166]. In vivo studies showed that CCP has a protective effect against the immunosuppression caused by cyclophosphamide. Treatment with CCP promoted the proliferation of lymphocytes induced by Con-A and LPS and promoted the activity of NK cells and macrophages in immunosuppressed mice. Mechanistic studies showed that CCP upregulates the expression of TLR4 and its downstream proteins [167].

The PPS polysaccharide from *Cordyceps gunnii* mycelia has antitumor effects in vitro [168] and immunostimulatory effects in vivo. In cyclophosphamide-treated, immunodeficient mice, the oral administration of PPS led to increased spleen and thymus indices, as well as NK cell cytotoxicity, and the proliferation of lymphocytes. The serum levels of IL-2, IL-12, IFN-γ, and IgG increased, while TGF-β levels decreased. Western blots showed that PPS acts via the TLR4/TRAF6/NF-κB signaling pathways [169].

The immunomodulatory activity of oyster polysaccharides was studied by Pan et al., in 2016. Among several subfractions, the C30–60% fraction showed the greatest positive effect on the proliferation of RAW264.7 cells and T lymphocytes in vitro. In mice xenograft tumor models, treatment with 5-FU, together with the polysaccharide, resulted in a higher tumor inhibition rate and increased the lymphocyte count, indicating the benefits of immune enhancement in the treatment of tumors. Composition analysis showed that the fraction contains a high ratio of uronic acid and sulfate content [170].

JNY2PW is a water-soluble polysaccharide isolated from *Arca inflata* that has a comb-like branched structure and a molecular weight of 5.25 × 10^4^ kDa. The treatment of RAW264.7 cells with JNY2PW promoted the expression of MHC II and CD86 on the cell surface, increased the production of inflammatory cytokines, and enhanced phagocytosis. The pretreatment of macrophages with a TLR4 inhibitor blocked the effects of JNY2PW, indicating that the activation of macrophages is mediated by TLR4. The activation of downstream signaling molecules, such as ERK, JNK, p38, and NF-κB, was also observed by Western blotting [171]. In an animal study, JNY2PW could alleviate the immunosuppression induced by cyclophosphamide and showed anticancer effects in xenograft tumor models [171].

In 2017, Sheng et al., found that the intragastric administration of *Hericium erinaceus* polysaccharide (HEP) significantly upregulates the immunity of mice intragastrically. In addition, the functions of NK cells and macrophages were enhanced, the level of hemolysin in the serum increased, and the proliferation of lymphocytes induced by mitogen was promoted. Western blotting showed that HEP affects the intestinal immune system by activating the AKT and MAPK signaling pathways. Thus, the immunomodulatory effects of HEP may occur via the activation of intestinal mucosal immunity [172].

A water-soluble polysaccharide GFP was isolated from the fruit of *Grifola frondosa* by Meng et al., in 2017, and it was found to activate RAW264.7 cells [173]. Further investigation in immunosuppressed animal models showed that GFP increases the spleen and thymus indices, as well as the serum levels of IL-2, IL-6, and TNF-α. NK cell cytotoxicity and lymphocyte proliferation were also enhanced by GFP. Western blotting showed that GFP upregulates JAK2/STAT3/SOCS signaling [174].

The immunomodulatory activity of a polysaccharide derived from longan *Dimocarpus longan* Lour. has also been investigated. The oral administration of the longan polysaccharide (LP) enhanced the immune function of immunosuppressed mice. A multi-omics-based network analysis revealed that LP administration changes the intestinal microbiome composition and metabolism (short-chain fatty acids) in mice [183]. LP also enhanced intestinal secretory IgA by upregulating TGFβRII, MAdCAM-1, and integrin α4β7 [175]. Together, these effects enhanced the immune system.

A galactomannan isolated from *Sesbania cannabina* [176] was shown to exhibit immunostimulatory activity. The treatment of RAW264.7 cells with the polysaccharide increased the levels of NO and intracellular ROS. The cytokines IL-1β, IL-6, IL-10, and IFN-γ all increased according to qRT-PCR, and higher expression of TLR2 and TLR4 was also induced by the polysaccharide [177].

Polysaccharides derived from alfalfa *Medicago sativa* L. also exhibit immunomodulatory bioactivities. In 2013, Li et al., reported that alfalfa polysaccharides (APS) have stimulatory effects on different immune cells, including splenetic lymphocytes, NK cells, DCs, and macrophages [184]. In 2019, Xie et al., found that the treatment of RAW264.7 cells with alfalfa polysaccharides increases the production and secretion of IL-6 and TNF-α. Western blotting and an inhibitor competition assay showed that alfalfa polysaccharides activate macrophages via the NF-κB and MAPK signaling pathways [178]. In 2020, the effects of alfalfa polysaccharides on B cells were reported, and it was shown that they increase the production of IgM and several cytokines by B cells. It was also shown that TLR4 is the primary receptor for the effects of alfalfa polysaccharides, which was confirmed by experiments on TLR4-deficient mice [179].

*Biophytum petersianum* is a traditional medicine used in West Africa. A polysaccharide of *B. petersianum*, BP100 III.1, has been shown to exhibit immunomodulatory activity. It was extracted with hot water and observed to possess potent activity in a complement fixation assay [180]. Another polysaccharide fraction, BPII, enhanced the immunological function of Peyer’s patch cells and macrophages [181]. BP1002 was found to activate macrophages and DCs but not T, B, or NK cells. The structure–activity relationship of BP1002 was explored by enzyme digestion experiments, which showed that rhamnogalacturonan type I is an important moiety for the immune enhancement activity [182]. 

## 4. Conclusions and Perspectives

The development of broad-spectrum antiviral drugs is important, especially for newly emerging viruses, such as COVID-19. Eliminating viruses by modulating the immunity of the host is a promising strategy that can be effective even without full knowledge of the specific virus, allowing it to theoretically lead to the discovery of broad-spectrum antivirals. Many natural products have shown antiviral effects mediated by enhancing immunity, which provides clues for the design of new drugs. Some natural products are known for their immunomodulatory activity, yet the antiviral effects remain unknown. There are still some obstacles to developing broad-spectrum antiviral drugs from natural products. For example, the targets and mechanisms underpinning some natural products remain unclear, potential side effects have not been fully investigated, and the activity of natural products is usually moderate. However, progress continues to be made in the study of natural products. Advanced technology for chemical modification, investigations into underlying biological mechanisms, and progress in drug design will facilitate future developments in the field. With the inspiration of natural products, enhancing immunity to overcome viruses may be possible, just as immunotherapy has already begun to show its strengths in oncology [185]. 

## Figures and Tables

**Figure 1 viruses-13-01257-f001:**
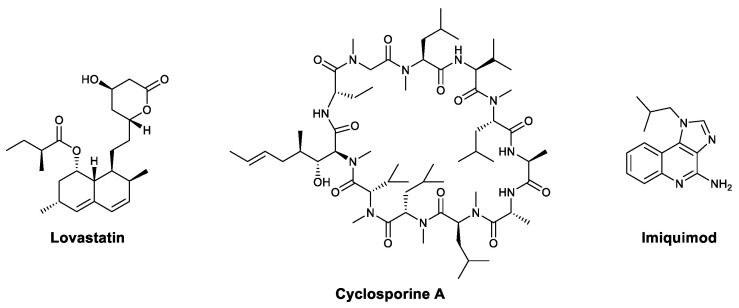
Structures of some broad-spectrum antiviral agents.

**Figure 2 viruses-13-01257-f002:**
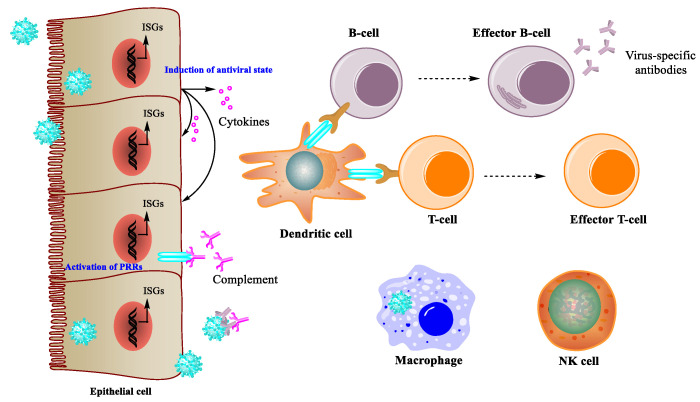
Immune response during viral infections.

**Figure 3 viruses-13-01257-f003:**
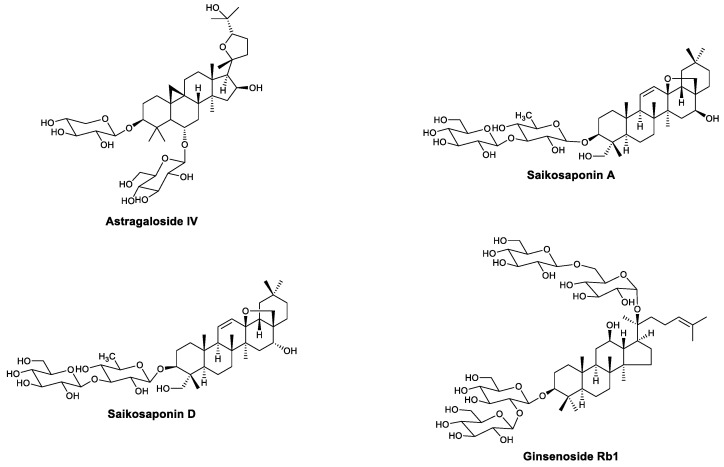
Structures of astragaloside IV, saikosaponins A and D, and ginsenoside Rb1.

**Figure 4 viruses-13-01257-f004:**
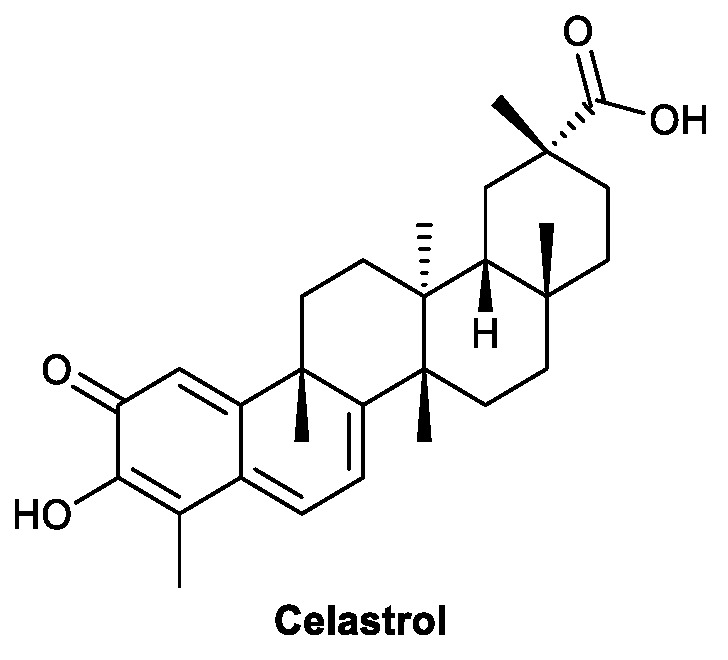
Structure of celastrol.

**Figure 5 viruses-13-01257-f005:**
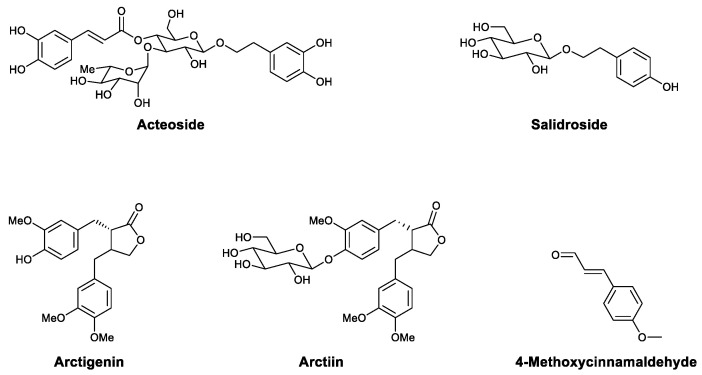
Structures of acteoside, salidroside, arctigenin, arctiin, and 4-methoxycinnamaldehyde.

**Figure 6 viruses-13-01257-f006:**
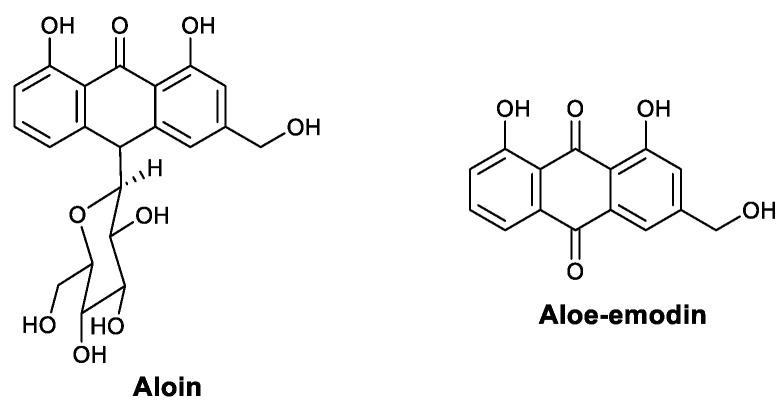
Structures of aloin and aloe-emodin.

**Figure 7 viruses-13-01257-f007:**
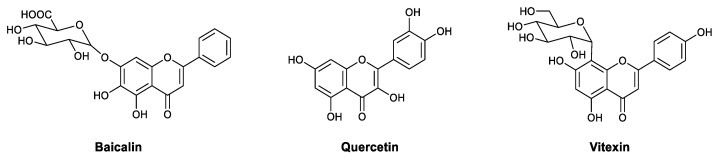
Structures of baicalin, quercetin, and vitexin.

**Figure 8 viruses-13-01257-f008:**
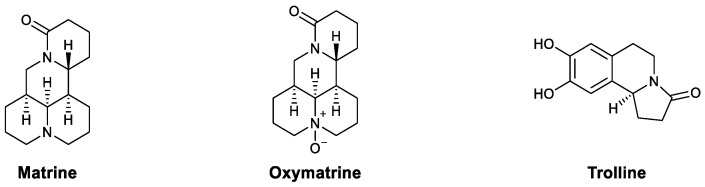
Structures of matrine, oxymatrine, and trolline.

**Figure 9 viruses-13-01257-f009:**
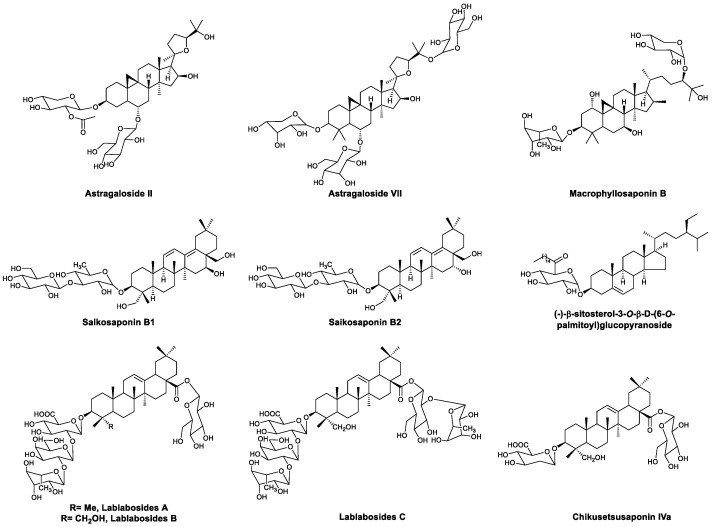
Structures of astragalosides II and VII; macrophyllosaponin B; saikosaponins B1 and B2; (−)-β-sitosterol-3-*O*-β-D-(6-*O*-palmitoyl)glucopyranoside; lablabosides A, B, and C; and chikusetsusaponin IVa.

**Figure 10 viruses-13-01257-f010:**
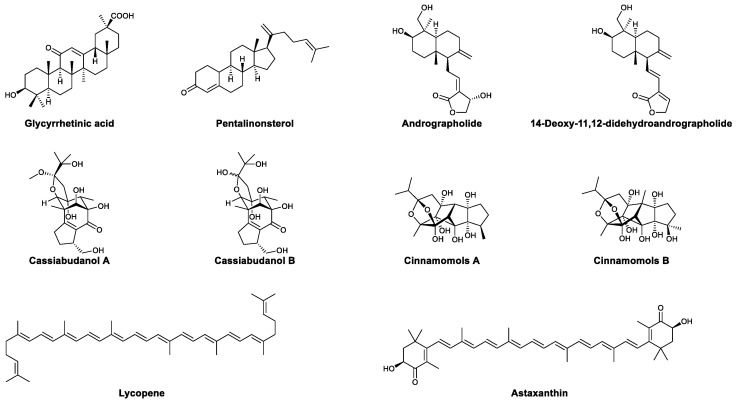
Structures of glycyrrhetinic acid, pentalinonsterol, andrographolide, 14-deoxy-11,12-didehydroandrographolide, cassiabudanols A and B, cinnamomols A and B, lycopene, and astaxanthin.

**Figure 11 viruses-13-01257-f011:**
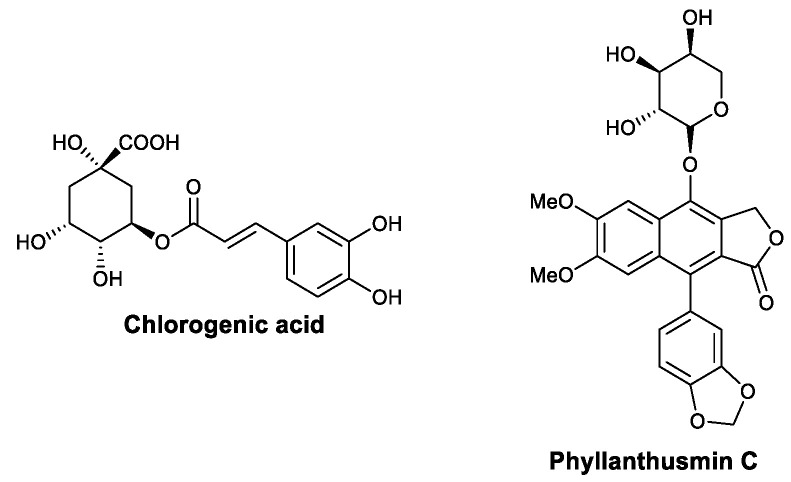
Structures of chlorogenic acid and phyllanthusmin C.

**Figure 12 viruses-13-01257-f012:**
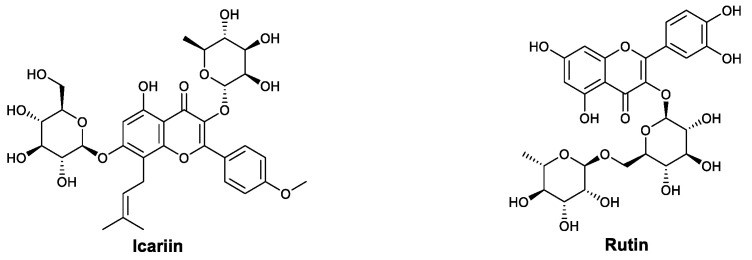
Structures of icariin and rutin.

**Figure 13 viruses-13-01257-f013:**
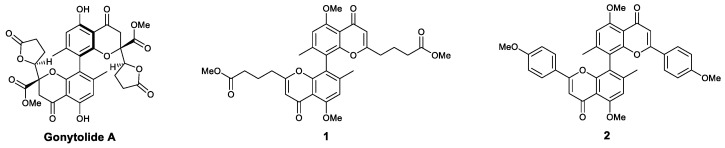
Structures of gonytolide A and its derivatives.

**Figure 14 viruses-13-01257-f014:**
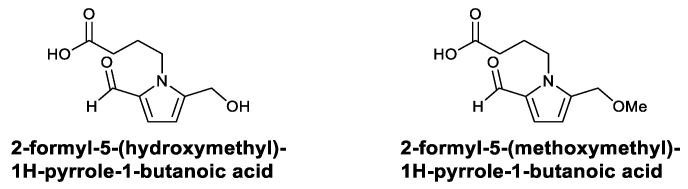
Structures of pyrrole alkaloids.

**Figure 15 viruses-13-01257-f015:**
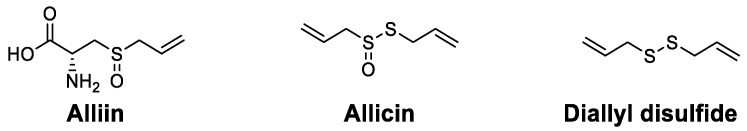
Structures of organosulfur compounds.

**Table 1 viruses-13-01257-t001:** Antiviral effects of natural immunomodulatory products.

Chemical Type	Compound Name	Antiviral Against	Mechanisms	Reference
Saponins	Astragaloside IV	CVB3	Upregulated IFN-γ, inhibited NF-κB	[30,31,32]
	Saikosaponin A	Influenza virus, PCV2	Downregulated NF-κB and promoted neutrophil and monocyte recruitment; upregulated IL-2 and IFN-γ and increased IgG and WBCs in serum	[33,34,35,36,37,38]
	Saikosaponin D	PCV2	Upregulated IL-2 and IFN-γ and increased IgG and WBCs in serum; activated macrophages	[35,36,37,38]
	Ginsenoside Rb1	EV71	Increased TNF-α, IL-10, IFNs, and antibodies in serum	[39,40]
Terpenoids	Celastrol	SARS-CoV, HIV, DENV	Activated the JAK/STAT pathway and increased IFN-α	[41,42,43]
Phenylpropanoids	Acteoside	VSV, influenza virus	Activated T-bet and increased IFN-γ; activated ERK to promote proliferation of lymphocytes	[44]
	Salidroside	CVB3, DENV, RSV	Increased IL-10 and IFN-γ, decreased TNF-α and IL-2; increased the expression of RIG-I and upregulated IRFs	[45,46]
	Arctigenin	H1N1, PCV2, JEV, HIV, HSV	Activated the TLR6/NOX2/MAPK signaling pathway and activated macrophages	[47,48,49,50,51,52,53]
	Arctiin	Influenza A	Increased antibody levels in serum	[48]
	4-Methoxy-cinnamaldehyde	RSV	Increased the secretion of IFN	[54]
Anthraquinones	Aloin	Influenza	Activated STAT, T-bet, and IFN-γ; increased the number of antigen-specific T cells	[55]
	Aloe-emodin	JEV, EV71	Increased IFN-α	[56]
Flavonoids	Baicalin	H1N1	Promoted the secretion of IFN-γ in T cells and NK cells	[57,58,59]
	Quercetin	HSV, influenza A, NDV, VSV	Increased type I IFN and IL-6	[60,61]
	Vitexin	Influenza virus	Upregulated TLR4 and increased IFN-β	[62]
Alkaloids	Matrine	PRRSV, PCV2	Enhanced phagocytosis and lymphocyte proliferation	[63]
	Oxymatrine	HBV	Upregulated the TLR9 pathway, promoted maturation of dendritic cells, and increased IFN-γ, TNF-α, and IL-2	[64,65,66]
	Trolline	Influenza virus	Upregulated TLR4 and increased IFN-β	[62]
Polysaccharides	GP	BVDB	Promoted the expression of IRFs	[67]
	RIP	HSV-2, influenza A, HBV	Activated the JAK/STAT pathway and upregulated IFN-α	[68,69,70]
	APS	Influenza A	Enhanced IL-1β, IL-10, IL-6, and TNF-α levels	[71]

**Table 2 viruses-13-01257-t002:** Small molecules that enhance immunity.

Chemical Type	Compound Names	Immunomodulatory Effects	Reference
Saponins	Astragaloside II	Increased IFN-γ and IL-2 and promoted the proliferation and activation of T cells	[73,74]
	Astragaloside VII, macrophyllosaponin B	Increased IL-2 and IFN-γ and promoted the proliferation of splenocytes	[75,76]
	Saikosaponins B1 and B2	Activated macrophages	[77,78]
	(−)-β-Sitosterol-3-*O*-β-D-(6-*O*-palmitoyl)glucopyranoside	Promoted the secretion of IFN-γ in NK cells	[79]
	Lablabosides A, B, and C	Adjuvant activity	[80]
	Chikusetsusaponin IVa	Adjuvant activity	[80]
Terpenoids	Glycyrrhetinic acid	Promoted the expression of TLRs and upregulated downstream MyD88, IL-6, and IFN-β; enhanced the proliferation of lymphocytes	[81,82]
	Pentalinonsterol	Activated macrophages and DCs, increased proinflammatory cytokines, and increased antibodies	[83,84]
	Andrographolide	Enhanced the formation of specific cytotoxic T lymphocytes	[85]
	14-Deoxy-11,12-didehydroandrographolide	Enhanced the innate immunity of intestinal epithelial cells	[86]
	Cassiabudanols A and B	Promoted the proliferation of splenocytes and increased the ratio of CD4+ and CD8+ T cells	[87]
	Cinnamomols A and B	Promoted the proliferation of T cells	[88]
	Lycopene	Increased the proportion of CD4+ and CD8+ T cells and increased IL-2, TNF-α, and IFN-γ	[89]
	Astaxanthin	Promoted the proliferation of lymphocytes, enhanced the activity of NK cells, and increased the production of antibodies	[90,91,92,93]
Phenylpropanoids	Chlorogenic acid	Activated macrophages via the calcineurin/NF-ATc/IL-2 pathway	[94,95]
	Phyllanthusmin C	Increased IFN-γ via the TLR/NF-κB pathway	[96]
Flavonoids	Icariin	Increased the production of antibodies, promoted the expression of TLR9 in macrophages, and downregulated inflammatory cytokines	[97,98,99,100,101,102,103]
	Rutin	Increased the population of CD4+ lymphocytes and promoted the secretion of IFN-γ	[104]
Chromanone	Gonytolide A	Activated innate immune system and increased the production of IL-8	[105,106]
the Alkaloids	2-Formyl-5-(hydroxymethyl)-1H-pyrrole-1-butanoic acid; 2-formyl-5-(methoxymethyl)-1H-pyrrole-1-butanoic acid	Activated macrophages and increased TNF-α, IL-12, and NO	[107]
Organosulfur compounds	Alliin	Promoted the proliferation of mononuclear cells and increased IL-1β	[108]
	Allicin	Increased proinflammatory cytokines and increased the populations of macrophages, DCs and CD4+ T cells	[109]
	Diallyl disulfide	Increased WBCs and antibodies and led to more effector B cells	[110]

**Table 3 viruses-13-01257-t003:** Chemical details of polysaccharides.

Plant	Polysaccharide	Molecular Weight	Constituents	Reference
*Phaseolus aureus*	Verbascose	0.83 kDa	Glu, Gal	[130]
	Acemannan	1.66 kDa	Man, Gal	[131,132,133,134,135,136]
*Aloe*	MAP	8 kDa	Man, Gal, Glu	[137]
	Aloeride	4000–7000 kDa	Glu (37.2%), Gal (23.9%), Man (19.5%), Ara (10.3%)	[138]
*Polygonatum sibiricum*	F1	103 kDa	Man, Glu, Gal, Ara	[139]
	F2	628 kDa		[139]
	PSP	-	Major: Gal, Rha Minor: Man, Glu, Xyl	[140]
*Astragalus membranaceus* (Fisch.) Bge.	-	1334 kDa	Rha, Ara, Glc, Gal, Gal (0.03:1.00:0.27:0.36:0.30)	[141]
	-	5.6 kDa	Glc, Gal, Ara, Xyl, Gal (10.0: 1.3: 1.7: 1.0: 0.9)	[142]
*Schisandra chinensis* (Turcz.) Baill	SCPP11	3.4 kDa	Man, Glu, Gal (1:11.38:3.55)	[143,144]
	Crude	-	Glu (38.0%), Gal (36.7%), galacturonic acid (12.0%), Ara (7.3%), Rha (4.0%), Man (1.2%), glucuronic acid (0.6%).	[145]
*Panax quinquefolium* L.	CVT-E002	-	Poly-furanosyl-pyranosyl-saccharides	[146,147]
	GL-4IIb2	11 kDa	15 different monosaccharides	[148,149,150]
*Bupleurum smithii* var. *parvifolium*	Crude	-	Major: Ara:Gal:Glu:Rha = 6.35:3.15:1.47:1 Minor: Man, Xyl	[151,152]
	D3-S1	2000 kDa	Major: Ara:Gal = 2.6:1.0 Minor: Rha, Glu, Xyl, Man	[153]
*Prunella vulgaris L*	PV2IV	-	Ara, Xyl, Man, Gal, Glu	[154,155]
	P1	-	Ara (28.37%), Xyl (54.67%), Man (5.61%), Glu (5.46%), Gal (5.89)	[156]
*Anemarrhena asphodeloides* Bunge	AS-1	-	Man, Gal, Ara, Glu	[157]
	AAP70-1	2.72 kDa	Glu, Fru	[158]
*Morus alba*	PMA	-	-	[159]
	Crude	-	Glu:Man:Ara:Gal:Xyl:Rha:ribose = 250:66:6:3.25:2.5:1.25:1	[160]
*Undaria pinnatifida*	Fucoidan	<10 kDa	Sugar (72.16% ± 0.31%), uronic acid (1.42% ± 0.03%), amino acids (7.04% ± 0.47%), sulfate (16.62% ± 1.31%)	[161]
*Ganoderma lucidum*	PS-G	-	Major: Glu, Man Minor: Fuc, N-acetylglucosamine, Xyl, Rha	[119]
*Platycladus orientalis (L.)* Franco	POP1	8.1 kDa	Rha (5.74%), Ara (12.58%), Man (10.97%), Glu (64.96%), Gal (6.55%)	[162]
*Antrodia cinnamomea*	ACP	70 kDa	-	[163]
*Phellinus sp.*	SHIP-1	-	Man:Glu:Gal:Ara:l-fucose = 1.92:1.00:2.37:0.44:1.13	[164]
	SHIP-2	-	Glu:Gal:l-fucose = 1.0:0.61:0.83	[164]
*Hordeum vulgare L.*	BP-1	67 kDa	-	[165]
*C. cornucopioides.*	CCP	1970 kDa	Man (48.73%), Gal (17.37%), Glu (15.97%), Xyl (17.93%)	[166,167]
*Cordyceps gunnii mycelia*	PPS	-	Man, Glu, Gal	[168,169]
*Oyster C. hongkongensis*	C30–60%	-	Glu (96.76%), Man (2.91%), Ara (0.24%), Ribose (0.04%), Gal (0.04%), Xyl (0.01%)	[170]
*Arca inflata*	JNY2PW	52500 kDa	-	[171]
*Hericium erinaceus*	HEP	-	Man (2.5%), glucuronic acid (1.1%), Glu (60.9%), Gal (28.0%), Fru (7.5%)	[172]
*Grifola Frondosa*	GFP	155 kDa	Rha:Xyl:Man:Glu = 1.00:1.04:1.11:6.21	[173,174]
Longan (*Dimocarpus longan* Lour.)	LP	377 kDa	Gal (40.88%), Ara (38.26%), Glu (9.00%), Rha (5.49%), Xyl (1.60%), ribose (1.05%), Fuc (0.91%)	[175,176]
Sesbania cannabina	Galactomannan	216 kDa	Gal:Man = 1.6:1	[176,177]
Medicago sativa L.	APS	-	Fuc, Ara, Gal, Glu, Xyl, Man, galacturonic acid, glucuronic acid	[178,179,180]
*Biophytum petersianum* Klotzsch	BP100 III.1	24 kDa	Ara:Rha:Fuc:Xyl:Man:Gal:GalA = 7.9:22.6:1.1:5.0:2.0:20.0:38.5	[180]
	BP1002	64 kDa	Rha (9.8%), Gal (10.5%), Ara (15.6%), Xyl (10.8%), galacturonic acid (45.8%)	[181]
	BPII	-	Major: GalA (65.1%) Minor: Gal, Rha, Xyl, Ara, Fuc, Glc, Man, GlcA	[182]

Abbreviations: Man, mannose; Glu, glucose; Gal, galactose; Ara, arabinose; Rha, rhamnose; Xyl, xylose; Fuc, fucose; Fru, fructose.

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
