# Peer review of "Development of Broad-Spectrum Antiviral Agents—Inspiration from Immunomodulatory Natural Products"

_viruses, 2021, doi:10.3390/v13071257_

Round 1

Reviewer 1 Report

The presented review work is devoted to the antiviral activity of natural compounds. The review is very well written and is of considerable scientific value. The importance of the presented work is high.

The main observation is that the title of the review is not correct. In my opinion, a review paper cannot be called "Discovery of ……". The authors should change the title. The discovery of anything can be based on your own results, not on an analysis of the literature

At the beginning of chapter 2.2 it makes sense to add a reference to a modern review on the antiviral activity of natural compounds. The review was published in 2021 and the authors may not have seen this work. O.I. Yarovaya, N.F. Salakhutdinov, "Mono- and sesquiterpenes as a starting platform for the development of antiviral drugs", RUSS. CHEM. REV., 2021, 90 (4), 488–510 DOI: https://doi.org/10.1070/RCR4969

Section 3.1 on the antiviral activity of low molecular weight compounds could be greatly expanded. Many of the natural agents with specific antiviral and immunomodulatory activity have been missed. For example, the works:

Yu J.S., Tseng C.K., Lin C.K., Hsu Y.C., Wu .H., Hsieh C.L., Lee J.C. Celastrol inhibits dengue virus replication via up-regulating type I interferon and downstream interferon-stimulated responses // Antiviral Res. -2017. –V. 137. P. 49-57. https://doi.org/10.1016/j.antiviral.2016.11.010

Song G., Yang S., Zhang W., Cao Y., Wang P., Ding N., Zhang Z., Guo Y., Li Y. Discovery of the First Series of Small Molecule H5N1 Entry Inhibitors // J. Med. Chem. -2009. –V. 52. -â„– 23. –P. 7368–7371.  http://pubs.acs.org/doi/abs/10.1021/jm900275m

There are many such examples and the authors should have clarified this point..

There are typos on the lines

181 – misspelling in Acteside

389, 789 - the beginning of the sentence should be capitalized

421, 482- no space between words

486 - an extra dot in the text

In some places the design of the reference list is given in a different style. Thus, page 18, the lines 721, 725 and 732. The authors need to bring uniformity.

753 missing a point

Error in depicting the structure Oxymatrine. Chemically, the structure should be + on nitrogen and - on oxygen

Overall, the review appears to be extremely useful for both virologists and specialists in medicinal chemistry.

Reviewer 2 Report

Zhang et. al. provide a review of natural products that have antiviral activity, primarily through modulation of the immune response.  This is a useful summary but there are issues that need to be addressed.

  1. The review just reads like a list.  This makes the review boring to read as it drones on and on.  It is also confusing as written.  The structure of the review is based on specific organisms where extracts or purified molecules have multiple effects.  So in one paragraph you read that extract or compound X increases cytokines, mediates T-cell proliferation, and up-regulates PRRs.  Then in the next paragraph you read that a different extract or compound has the same or very similar effects. tthis makes it difficult to see the current state of the field at the 10,000 foot level.  Are there more general conclusions that could be made or can some overall synthesis of the field be presented?  This would improve the manuscript.  Perhaps more tables would help.  One way to address the could be a table that lists compound activities based on specific effects on immune enhancement.  For example, list TLR activation and then under that list the compounds shown to have that effect.
  2. There are numerous errors in English usage that need to be corrected.  For example, since they are reviewing previous work, the article should be written in past tense.  They actually mix tenses in the paper.  There is also a lack of use of definite articles.
  3. The authors don't include a discussion of the potential downsides of increasing immune function in the context of a viral infection and this could be an issue.  For example, in the current COVID19 pandemic it is becoming clear that some people who have been hospitalized have a hyperactive immune response that increases disease severity.
  4. The authors also include some generalizations that are overbroad or incorrect.  For example in lines 64 and 65 they state that "smallpox and polio virus have been eradicated from the world?  This is true for smallpox but polio still circulates in seven or so countries in the world so it has not been eradicated.

Reviewer 3 Report

The manuscript “Discovery of Broad-Spectrum Antiviral Agents from Immuno-modulatory Natural Products” by Zhang M. et al. reviews the current state-of-the-art of the research regarding broad-spectrum antiviral natural products discovery with particular focus on molecules and extracts with immunomodulatory functions, as promising starting points for antiviral drugs development. The review deals with an interesting important and current topic. Emerging and re-emerging viruses, which have resulted in many outbreaks, epidemics, and pandemics, including the current COVID-19 pandemic disease, have indeed highlighted the need of identifying  novel drugs with broad-spectrum antiviral activity; in this context natural products play a key role as a rich source of novel chemical compounds including antivirals .

The authors reviewed most of the literature  available on  the antiviral and immunomodulatory activity of natural products and extracts. However the English language and words are often not correctly used and need extensive revision. In my opinion the manuscript need major revision, as indicated, and resubmission.

Specific comments:

  1. English style is a major concern of the whole manuscript, thus revision of the language is mandatory for acceptance.
  2. Some aspects should be improved to make the manuscript more fluent and understandable. For example, the first part (paragraphs 2.2 and 2.3) is organized as a list of natural compounds and affected targets, lacking a sequential and smooth description. In order to provide a more integrated and intelligible description, one or more tables or figures schematizing the evidences linking   selected natural products with their principal targets and their antiviral activity spectra would help comprehension and reading of the review.
  3. Table 1 is not mentioned in the text.

Round 2

Reviewer 2 Report

The review has been improved by the addition of tables summarizing the activities but the writing is still turgid and doesn't flow well.  Technically the English is fine but the article is dense.

Reviewer 3 Report

The authors replied in an exhaustive manner to almost all of my observations. Two more tables  summarizing antiviral activities of natural products have been added, providing a more intelligible description.  

However, the manuscript still requires careful editing (for exampe  line 74 : yhe word "for" should be replaced with "from"; lines 90-92: the verb is missing in the sentence), after which I believe it can be accepted for pubblication